# Optimizing the parameterization of deep mixing and internal seiches in one-dimensional hydrodynamic models: a case study with Simstrat v1.3

Adrien Gaudard[1], Robert Schwefel[2], Love Råman Vinnå[2], Martin Schmid[1], Alfred Wüest[1,2] and Damien Bouffard[1,2]

[1]Eawag, Swiss Federal Institute of Aquatic Science and Technology, Surface Waters – Research and Management, Seestrasse 79, CH-6047 Kastanienbaum, Switzerland
[2]École Polytechnique Fédérale de Lausanne, Physics of Aquatic Systems Laboratory – Margaretha Kamprad Chair, EPFL-ENAC-IIE-APHYS, CH-1015 Lausanne, Switzerland

*Correspondence to*: Adrien Gaudard (adrien.gaudard@eawag.ch) or Damien Bouffard (damien.bouffard@eawag.ch)

**Abstract.** This paper presents an improvement of a one-dimensional lake hydrodynamic model (Simstrat) to characterize the vertical thermal structure of deep lakes. Using physically based arguments, we refine the transfer of wind energy to basin-scale internal waves (BSIWs). Namely, we consider the properties of the basin, the characteristics of the wind time series and the stability of the water column to filter and thereby optimize the magnitude of wind energy transferred to BSIWs. We show that this filtering procedure can significantly improve the accuracy of modelled temperatures, especially in the deep water of lakes such as Lake Geneva, for which the root mean square error between observed and simulated temperatures was reduced by up to 40 %. The modification, tested on four different lakes, increases model accuracy and contributes to a significantly better reproduction of seasonal deep convective mixing, a fundamental parameter for biogeochemical processes such as oxygen depletion. It also improves modelling over long time series for the purpose of climate change studies.

## 1 Introduction

### 1.1 Hydrodynamics of vertical mixing in lakes

Lakes are recognized as sentinels of changes in climate and catchment processes (Shimoda et al., 2011; Adrian et al., 2009). They have multiple intricate interactions with their environment, simultaneously reacting to external forces and acting on their surroundings. The complex hydrodynamic processes occurring in stratified lakes are mainly governed by the combination of surface heat flux and wind stress (Bouffard and Boegman, 2012). The former sets up a density stratification by warming the near-surface water, which floats on top of the cold deep water. This stratification pattern isolates the lower parts of the lake (hypolimnion) from the surface layer (epilimnion) and acts as a physical barrier reducing vertical fluxes. The latter, wind stress, brings momentum into the system and thereby contributes to mixing. Notably, the action of wind stress on a stratified basin leads to internal waves (seiches), rerouting the energy at various spatial and temporal scales (Wüest and Lorke, 2003; Wiegand and Carmack, 1986). Basin-scale internal waves (hereafter BSIWs) play a crucial role in

the transport of mass and momentum in the lake, driving horizontal dispersion and vertical mixing with important implications for biogeochemical processes (Bouffard et al., 2013; Umlauf and Lemmin, 2005).

In situ measurements, laboratory experiments and theoretical considerations have shown that the response of a stratified basin to wind depends on the strength, duration and homogeneity of the wind field as well as the geometry of the basin and the stratification of the water column (Valerio et al., 2017; Valipour et al., 2015; Stevens and Imberger, 1996; Mortimer, 1974). The typical assumption consists in considering the stratified water body as a two-layer system with different densities and thicknesses. The wave period of longitudinal or transversal standing waves ($T_{BSIW}$ [s]) can then be estimated with the Merian formula (Bäuerle, 1994; Merian, 1828):

$$T_{\mathrm{BSIW}} = 2L \left( n^2 g \frac{\rho_2 - \rho_1}{\rho_2} \frac{h_1 h_2}{h_1 + h_2} \right)^{-1/2} \qquad (1)$$

where $L$ [m] is the length of the basin at the interface depth (in the direction of the excitation), $g$ [m s$^{-2}$] is the acceleration of gravity, $\rho_1$ and $\rho_2$ [kg m$^{-3}$] are the densities of the upper and lower layers, $h_1$ and $h_2$ [m] are the layer thicknesses and $n$ is the number of wave nodes.

In most lakes, the zone of abrupt temperature change, commonly referred to as thermocline or metalimnion, progressively deepens from spring to autumn. The temperature difference between the two layers, determining the stratification strength, decreases from mid-summer to autumn. The net result of both a deepening of the thermocline and a reduced stratification strength is a strong increase in $T_{BSIW}$ (Eq. (1)). This effect is remarkably strong in meromictic and oligomictic lakes, which can maintain stratification throughout the whole year. Lake Geneva can be categorized as oligomictic with a period of the dominant first-mode longitudinal BSIW ranging from ~60 to ~600 h, as shown in Fig. 1. As expected, the wave period greatly increases from late autumn to early spring due to weakening of the stratification.

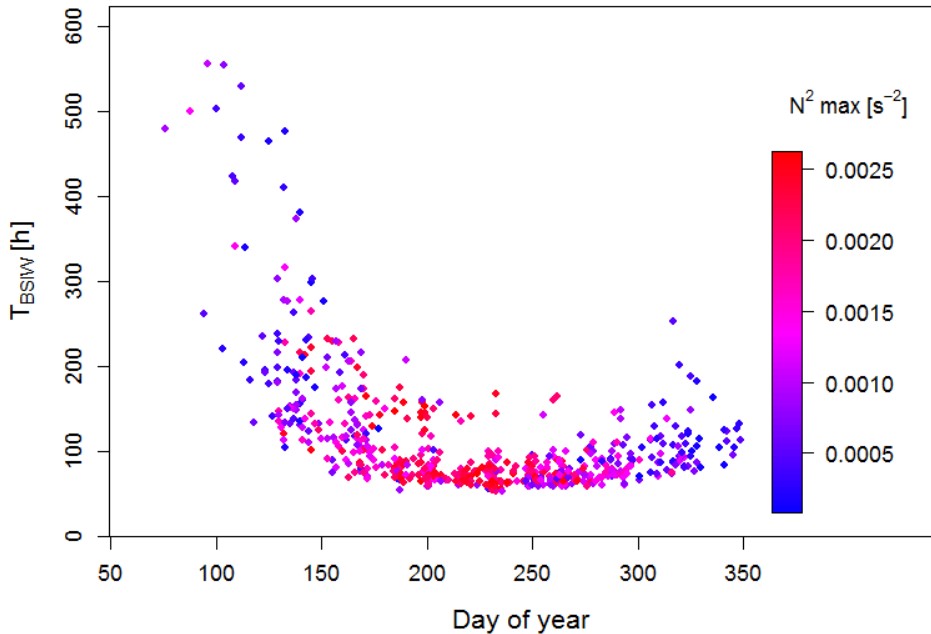

**Figure 1 – Period of the first longitudinal mode BSIW as a function of the day of year in Lake Geneva. Points are coloured based on the observed maximum water column stability $N^2$. Values of $N^2 < 10^{-4}\,s^{-2}$ correspond to rare instances of complete lake overturn, i.e. unstratified water column, and are consequently not shown. The figure was constructed from more than 50 years of monitoring data at the deepest location of the lake.**

Besides wind-induced BSIWs, convection is a subtle although important vertical energetic pathway in lakes (Read et al., 2012; Wüest et al., 2000). This process may be triggered by surface cooling or density currents (Thorpe et al., 1999). The generally weak winds over perialpine lakes increase the relative importance of convective processes in these systems. Notably, cooling during winter is responsible for temperature and thus density homogenization in mid-latitude water bodies.

In deep lakes however, the vertical mixing caused by convective and wind-induced turbulent processes (Imboden and Wüest, 1995) may be insufficient to mix the water column down to the deepest layers. In this case, the deepest part of the lake remains separated from the homogenized upper part (Jankowski et al., 2006; Straile et al., 2003). The extent of deep mixing is of critical importance for reoxygenation of the deep water, and therefore for water quality. Insufficient oxygen supply often affects lake ecosystems and can trigger the release of harmful compounds and reduced substances as iron or manganese (Friedrich et al., 2014; Beutel and Horne, 1999). Comprehension and accurate prediction of the extent of deep mixing are thus vital for proper management of water resources, especially in deep lakes. Moreover deep mixing, as a key process of heat transfer to the hypolimnion, partly governs its response to climate change (Ambrosetti and Barbanti, 1999). However, the complex vertical pathway of kinetic energy makes modelling of the thermal structure challenging.

## 1.2 Numerical modelling

Over the past decades, many numerical models have been developed to allow for prediction and understanding of hydrodynamics in lakes and reservoirs. The algorithms and assumptions applied by these models vary greatly, as does their

complexity, which ranges from simple box models to intricate three-dimensional models (Stepanenko et al., 2016; Wang et al., 2016; Ji, 2008; Hodges et al., 2000). For climate change studies, long-time lake simulations are needed, often utilizing vertical one-dimensional (1D) models (Wood et al., 2016; Butcher et al., 2015; Komatsu et al., 2007; Hostetler and Small, 1999). These models are computationally inexpensive (Goyette and Perroud, 2012), permitting long-term simulations

(several decades or longer) and facilitating parameter estimation and sensitivity analyses.

Common 1D lake models have been extensively studied, validated and compared with different sets of observational data (Thiery et al., 2014; Perroud et al., 2009; Boyce et al., 1993). They have been proven to satisfactorily simulate the seasonality of surface temperature. Still, problems arise in reproducing the vertical mixing through the thermocline and the evolution of the deep water temperature (Schwefel et al., 2016; Stepanenko et al., 2010). Lake models have often been

developed for specific applications and do not explicitly include all relevant physical processes and interactions. For instance, while processes like shear-induced turbulent mixing and solar radiation are usually modelled, the effect of internal waves is commonly not parameterized (Stepanenko et al., 2010). As a result, the estimation of bottom temperature appears to be significantly less accurate than that of surface temperature (Goyette and Perroud, 2012). This leads to large discrepancies between different models under similar running conditions (Stepanenko et al., 2013). Model accuracy can be improved by

including BSIW parameterization and subsequent mixing (Schwefel et al., 2016). The problem has been partly tackled for small basins (Stepanenko et al., 2014), but not for large, deep lakes.

An inherent weakness of 1D models is the neglect of horizontal processes. These models may be able to reproduce fine vertical structures but generally perform calculations that are horizontally averaged over the whole simulation domain (Komatsu et al., 2007). Beside the classical problem of spatial heterogeneity, the ratio between vertical and horizontal

turbulent viscosity, recognized to impact the response of lakes to wind forcing (Toffolon and Rizzi, 2009), cannot be accounted for. When vertical and horizontal viscosity are on the same order of magnitude (typically in winter), horizontal circulation is favoured and wind energy will give rise to large horizontal gyres rather than vertical motion (Straile et al., 2010). The stratified situation, associated with smaller vertical viscosity in comparison to horizontal viscosity, will favour BSIWs as well as up- and downwelling, the energy being absorbed by thermocline displacement and subsequently dissipated

(Woolway and Simpson, 2017). We hypothesize that in 1D models resolving BSIWs, too much wind energy is introduced and converted into BSIW in winter, incorrectly impacting deep convection patterns (Fink et al., 2014b).

Here, we propose physically motivated modifications of the BSIW parameterization of a well-established 1D numerical model for better reproduction of deep mixing and hence improved prediction of deep water temperature. In Section 2, we present the model, describe our modifications and introduce the sites for which we test the model. In Section 3, we discuss

some aspects of our modifications and compare both calibration and results of the initial and modified models. Finally, we discuss the implications of our improvements.

## 2 Methods

In this study, we use the one-dimensional numerical model Simstrat v1.3, a modified version of the model developed by Goudsmit et al. (2002). Simstrat is a finite-difference buoyancy-extended reservoir model with k-ε turbulence closure. It has been extensively used to simulate water column temperature development under different conditions and over periods ranging from days to decades (Schwefel et al., 2016; Fink et al., 2014b; Stepanenko et al., 2014; Straile et al., 2010; Peeters et al., 2007, 2002). The model performs equally good and in many aspects even better than other models, as shown in a comparison study simulating Lake Geneva (Perroud et al., 2009). A possible reason for that is the implemented explicit parameterization of turbulence caused by BSIWs through the partitioning of wind energy.

### 2.1 Parameterization of internal waves

Simstrat parameterizes the total BSIW energy $E_{\text{seiche}}$ [J] as a balance between production $P_{\text{seiche}}$ [W] and loss $L_{\text{seiche}}$ [W] of seiche energy (Goudsmit et al., 2002):

$$\frac{dE_{\text{seiche}}}{dt} = P_{\text{seiche}} - L_{\text{seiche}} \tag{2}$$

Within the standard system of equations of a k-ε model, where k is the turbulent kinetic energy and ε its dissipation, $P_{\text{seiche}}$ is used as a source term in the differential equation for k and, multiplied by ε/k and a constant, as a source term in the differential equation for ε. $P_{\text{seiche}}$ is parameterized as a function of the energy introduced by wind forcing at the lake surface:

$$P_{\text{seiche}} = \alpha A_{\text{lake}} \rho_{\text{air}} C_{10} U_{10}^3 \tag{3}$$

where $\alpha$ [-] is the efficiency of energy conversion from wind to seiches, $A_{\text{lake}}$ [m$^2$] the surface area of the lake, $\rho_{\text{air}}$ [kg m$^{-3}$] the density of air, $C_{10}$ [-] the wind drag coefficient and $U_{10}$ [m s$^{-1}$] the wind speed 10 m above the water surface. The following power law is used to calculate $L_{\text{seiche}}$ from the total seiche energy $E_{\text{seiche}}$:

$$L_{\text{seiche}} = A_{\text{lake}} V_{\text{lake}}^{-3/2} \rho_{\text{water}}^{-1/2} C_{\text{Deff}} E_{\text{seiche}}^{3/2} \tag{4}$$

where $V_{\text{lake}}$ [m$^3$] is the volume of the lake, $\rho_{\text{water}}$ [kg m$^{-3}$] the density of water and $C_{\text{Deff}}$ [-] the bottom drag coefficient.

The wind drag coefficient is calculated according to an empirical relation based on Wüest and Lorke (2003) – which is not the original formulation of Goudsmit et al. (2002):

$$C_{10} = \begin{cases} 6.215 \times 10^{-2} & ; \quad U_{10} \leq 0.10 \text{ m s}^{-1} \\ 4.4 \times 10^{-2}\, U_{10}^{-1.15} & ; \quad 0.10 < U_{10} \leq 3.85 \text{ m s}^{-1} \\ -7.12 \times 10^{-7} U_{10}^2 + 7.387 \times 10^{-5}\, U_{10} + 6.605 \times 10^{-4} & ; \quad U_{10} > 3.85 \text{ m s}^{-1} \end{cases} \tag{5}$$

The modelling choices of Simstrat have been proven to yield very good estimates of water column temperature during the strongly stratified summer conditions in Lake Geneva (Perroud et al., 2009). However, the model was shown to invariably overestimate the intensity of winter deep mixing in Lake Geneva. Therefore, it was unable to follow deep water temperature evolution over long timescales. The turnover of the lake is significantly exaggerated by the model, yielding more unstable

deep water temperature than observed. An arguable solution to this problem would be prioritizing the deep water temperature during model calibration. While doing so, however, the wind-driven mixing observed in the thermocline region is then insufficiently represented (Stepanenko et al., 2010). For Lake Geneva, this disequilibrium is exemplified in Fig. 2 (upper frame) comparing observed and modelled temperature at 50 and 250 m depth. Mixing is clearly underestimated at a depth of 50 m and overestimated at a depth of 250 m. As a result, in most winters the simulated mixing depth strongly exceeds the observed value (Fig. 2, lower frame).

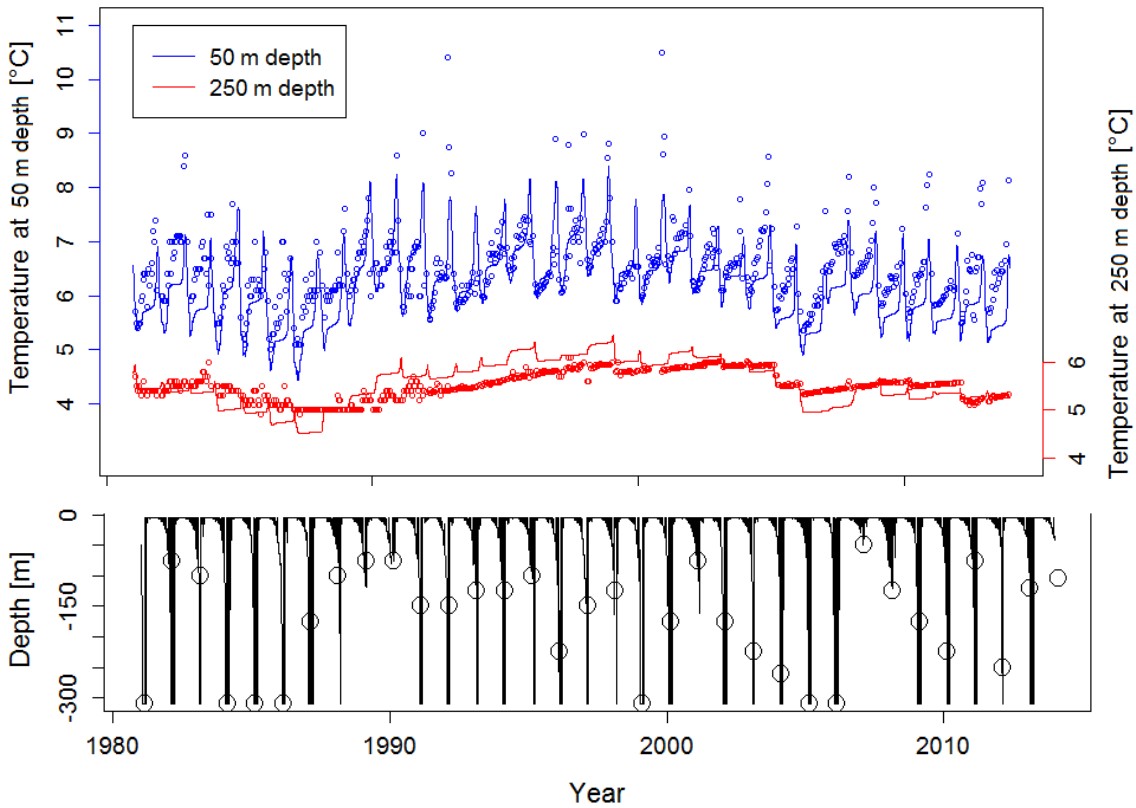

**Figure 2 – Upper frame: evolution of observed and modelled temperature for Lake Geneva. Lower frame: evolution of observed and modelled mixing depth for Lake Geneva. Circles represent observations, lines represent results obtained with the unmodified Simstrat model. The mixing depth is defined as the uppermost depth at which the absolute temperature gradient exceeds 0.005 °C m$^{-1}$, using time-averaged temperature time series over 60 days.**

## 2.2 Proposed improvements

Our approach consists in parameterizing the energy pathway from wind to BSIWs with two factors representing different controlling processes. Those factors are used to calculate a filtered wind speed $U_{10}^{\text{filtered}}$ for subsequent estimation of the BSIW energy production in Simstrat with Eq. (3). The focus here is not a global reduction of wind strength, but rather filtering at the event-scale as a function of the present state of the lake and of wind forcing. The standard wind speed $U_{10}$ is still used for all other calculations, i.e. shear stress and heat flux.

$$U_{10}^{\text{filtered}} = f_{\text{dur}} f_{\text{stab}} U_{10} \tag{6}$$

$f_{\text{dur}}$ and $f_{\text{stab}}$ range from zero to one and consequently induce a reduction of the effective wind speed available for BSIW energy production. We filter the wind based on the duration of wind events with $f_{\text{dur}}$ [-] and based on lake stability with $f_{\text{stab}}$ [-].

Regardless of their intensity, wind events much shorter than the BSIW period (from Eq. (1)) will inefficiently excite the oscillation modes of the internal waves. Specifically, wind stress lasting less than one fourth of the wave period should be filtered out (Schwefel et al., 2016; Stevens and Imberger, 1996). This limitation is parameterized in $f_{\text{dur}}$ and is computed as follows:

$$f_{\text{dur}} = \min\left\{ \left( \frac{T_{\text{wind}}}{T_{\text{BSIW}}/4} \right)^{1/2}, 1 \right\} \tag{7}$$

Here $T_{\text{wind}}$ [s] is the duration of a wind event and $T_{\text{BSIW}}$ [s] is the period of the first horizontal mode of BSIWs given in Eq. (1). The ratio is raised to the power of one half in order to smooth the cut-off effect induced by the chosen criterion (one-fourth of the BSIW period). Wind events are defined as the periods with mean wind speed at least equal to 1.5 times the overall average. Their duration is then the longest interval such that this condition is satisfied. The filter is applied at all times during the whole duration of the corresponding wind events. This filter is simplified since it considers neither the resonance effects nor the other oscillation modes of a given basin. However, we hypothesize that despite its simplicity, the filter covers most of the BSIW excitation range for large lakes.

The stability of the water column and the depth of the mixed layer both oppose excitation of BSIWs. We incorporate this in $f_{\text{stab}}$ by utilizing the Wedderburn number $W$ [-] (Imberger and Hamblin, 1982). $W$ measures the ratio between buoyancy force and wind stress, and scales inversely to the magnitude of vertical mixing in stratified basins (Shintani et al., 2010). $W$ is defined as:

$$W = g \frac{\rho_2 - \rho_1}{\rho_1} \frac{h_1^2}{u_*^2 L_{\max}} \tag{8}$$

Here $\rho_1$, $\rho_2$ [kg m$^{-3}$] the densities of the upper and lower layers, $h_1$ [m] the thickness of the upper layer, $L_{\max}$ [m] the length of the basin at the surface and $u_*$ [m s$^{-1}$] is the water-side shear velocity:

$$u_*^2 \approx \frac{C_{10} U_{10}^2 \rho_{\text{air}}}{\rho_1} \tag{9}$$

The reduction factor based on the stability of the water column is then computed as:

$$f_{\text{stab}} = \frac{1}{1 + \dfrac{W}{L/(4h_1)}} \tag{10}$$

Here $L$ [m] is the approximate length of the basin at the interface depth (see Eq. (11)) and $h_1$ [m] is the thickness of the upper layer. The normalizing factor $L/(4h_1)$ was discussed as an appropriate threshold for the amplitude of BSIWs (Adams and Charles, 2000). Both $L$ and $W$ are bounded from below to their respective minimal non-zero value. This avoids computation errors and total filtering of the wind time series when the lake is well mixed (therefore consisting in a single layer of thickness $h_1$, with $L = 0$ and $W = 0$; in this case the lower layer is defined using the bottom temperature).

Arguably, $f_{dur}$ and $f_{stab}$ build on similar principles. However, they incorporate different and complementary filtering of the wind field. $f_{dur}$ explicitly takes the duration of wind events into account, and therefore the physical resonance effects inherent to wind excitation of a water body. $f_{stab}$ compares the actual wind stress at the water surface to the overall stability of the water column.

The construction of the filtered wind requires knowledge of the wind characteristics before and after the present time, as well as of the corresponding state of the lake at each time step. Therefore, the model should be run with the unfiltered wind speed in a first step. Afterwards, the filter can be applied using the output of the first model run to calculate the filter parameters and determine $U_{10}^{filtered}$ from Eq. (6) for a second run. An improvement would be to calculate wind filtering at runtime (i.e. during model execution).

Verification of the wind filtering algorithm was performed (see "Code and data availability" Section).

## 2.3 Sites presentation and in situ data

The proposed change to the modelling scheme is applied and tested on four perialpine lakes in Switzerland. The chosen lakes range in volume from 1.1 to 89 km$^3$, representing the general conditions for medium to large-sized lakes. Information for each lake is provided hereunder. Information about the data sources is provided in the "Code and data availability" Section.

**Lake Geneva** (46.452° N, 6.620° E) is the largest lake of western Europe, located between France and Switzerland. Overall, the lake has a maximum length of ~75 km, a maximum width of ~13 km and a surface area of 580 km$^2$. The maximum depth is 309 m and the average hydraulic residence time is about 11 years. The Rhône River is both the main inflow and outflow. The lake is heavily affected by wind-induced currents, mixing and internal waves (Bouffard and Lemmin, 2013). Concurrently, its size makes it thermally stable, and the mild climate maintains a relatively high hypolimnion temperature (above 5 °C) (Michalski and Lemmin, 1995). Full mixing occurs irregularly in Lake Geneva (every 7 years on average).

Sixty years of vertical temperature profiles, obtained with a conductivity-temperature-depth probe (CTD), are available at the deepest point. This combined dataset is used for model initialization and calibration. The weather station in Pully (46.512° N, 6.667° E) provides meteorological data necessary for model forcing since 1981: air temperature, humidity, wind speed and direction, solar radiation and cloudiness. Given the large retention time of the basin, we neglect river inflow and outflow.

**Lake Constance** (47.629° N, 9.374° E) is a deep perialpine lake with a maximum depth of 254 m. The Rhine River acts as its main tributary, resulting in a hydraulic residence time of about 4.5 years. The maximum length is ~50 km and the width ~12 km, for a surface area of 536 km$^2$. Seasonal mixing reaches the lake bottom every few years (Fink et al., 2014a).

Surface forcing data is provided by the weather station in Güttingen (47.602°N, 9.279°E) since 1981, while CTD profiles and hydrological data about the inflow rate and temperature are available since 1984. Both inflow and outflow are assumed to take place near the lake surface. Monthly CTD measurements at the deepest point cover the complete time period.

**Lake Neuchâtel** (46.904° N, 6.843° E) is a medium-sized lake located in western Switzerland. Although relatively deep (maximum 152 m), Lake Neuchâtel is monomictic. It is therefore a good test case for our numerical model scheme to assess the correct reproduction of deep mixing. Its surface area is 218 km$^2$ and the average hydraulic residence time is slightly more than 8 years.

Meteorological data are available since more than 35 years from the stations listed in Table 1, but regular CTD profiling started only in 1994. Given the long residence time, inflows and outflows are neglected.

**Lake Biel** (47.104° N, 7.798° E) is the smallest and shallowest of the lakes analysed in this study. With a surface area of ~40 km$^2$ and a maximum depth of 74 m, it is located directly downstream (i.e. to the North-East) of Lake Neuchâtel. It is also monomictic. Since the correction of the Aare River in the 19th century, Lake Biel has a very short average retention time of only 60 days (Albrecht et al., 1999). Inflows thus play a primary role in the hydrodynamic behaviour of the lake.

As for Lake Neuchâtel, the CTD measurement program started in 1994. The forcing time series for Lake Biel was likewise computed based on several meteorological stations around the lake, listed in Table 1. For the inflows, a gravity-driven intrusion algorithm adapted from Hipsey et al. (2013) is applied, with an inflow plunging in the lake, entraining water and stabilizing when density equilibrium is reached. The inflow corresponds to the sum of the Hagneck River, the Suze River and the Zihl channel, and density effects of salinity and suspended matter are neglected.

The most important aspects of these four lakes are summarized in Table 1. For Lake Neuchâtel and Lake Biel, several weather stations are used to fill data gaps in the prioritized measurement station. In this case, missing data were completed with data from nearby weather stations, which were linearly adjusted to have the same mean as the priority station.

| | Max. depth | Area | Grid layers | Simulation period | Weather station(s) (first is prioritized) |
|---|---|---|---|---|---|
| Lake Geneva | 309 m | 580 km$^2$ | 600 | 01.01.1981 - 01.01.2014 | Pully |
| Lake Constance | 254 m | 536 km$^2$ | 600 | 01.01.1984 - 01.01.2012 | Güttingen |
| Lake Neuchâtel | 152 m | 218 km$^2$ | 450 | 01.01.1994 - 01.01.2015 | Neuchâtel, Bullet / La Frétaz, Chaumont, Cressier, Mathod, Mühleberg, Payerne |
| Lake Biel | 74 m | 40 km$^2$ | 300 | 01.03.1994 - 01.01.2015 | Cressier, Biel/Bienne, Mühleberg, Chasseral, Neuchâtel |

**Table 1 – Main properties of the study sites.**

## 2.4 **Model configuration and calibration**

The model is set to use a time step of 300 seconds. Both the initial model and the improved version are calibrated with the PEST (Model-Independent Parameter Estimation and Uncertainty Analysis) software package (Doherty, 2005), using its default setup. PEST searches for the optimal value of chosen parameters which ensure the best match between model results and available observations (here, temperature measurements over the simulation period). Only three parameters are chosen for calibration: the fraction of wind energy converted to seiche energy (α, see Eq. (3)), the fit parameter for absorption of longwave radiation ($p_1$) and the fit parameter for the fluxes of sensible and latent heat ($p_2$). $p_1$ linearly scales the amount of heat that is absorbed in the lake water from the incoming longwave radiation from the atmosphere, and $p_2$ linearly scales the exchange of sensible and latent heat between the lake surface and the atmosphere (Goudsmit et al., 2002). $p_1$ and $p_2$ account for the fact that, in specific cases, there is generally a bias between the calculated and the effective heat fluxes, as empirical formulas are used and the meteorological data is not always fully representative for the lake surface. All the other parameters are set to standard values (see Table 2), which makes the optimization procedure more solid and more trustworthy. Latitude is set according the above description of the sites, and the geothermal heat flux is set as follows: 0.50 W m$^{-2}$ for Lake Geneva, 0.00 W m$^{-2}$ for Lake Constance, 0.15 W m$^{-2}$ for Lake Neuchâtel and 0.08 W m$^{-2}$ for Lake Biel. These values match the temperature slopes found in the deep water over summer and autumn (when heat input from above is negligible). Deep river inflows are not modelled, although they can bring heat to the hypolimnion in summer (Fink et al., 2016).

As mentioned above, the focus of the wind filter is not a global reduction of the energy input to BSIWs. Instead, it should act at the event-scale, depending on the current state of the lake and wind forcing. The parameter α then allows to adjust the absolute magnitude of the energy input.

| Parameter description and units | Parameter in Goudsmit et al. (2002) | Value |
|---|---|---|
| Geographical latitude [°] | − | Lake-dependent |
| Air pressure [mbar] | $p_a$ | 990 |
| Fractionation coefficient for seiche energy [-] | $q$ | 1.25 |
| Bottom drag coefficient [-] | $C_{Deff}$ | 0.002 |
| Geothermal heat flux [W m$^{-2}$] | $H_{geo}$ | Lake-dependent |
| Fraction of short-wave radiation absorbed as heat in the uppermost layer [-] | − | 0.30 |

**Table 2 – Fixed (non-calibrated) physical parameters of the model.**

The length of each lake, used to compute BSIW period, was calculated with the surface area of the lake $A_{lake}$ according to:

$$L = 2\sqrt{A_{lake}} \tag{11}$$

This rather simplistic formula assumes a rectangular shape with aspect ratio equal to four, but limits the number of input parameters needed by the model. $L$ is then the fetch along the longest dimension of the basin.

## 3 Results and discussion

### 3.1 Wind filtering

A comparison of the filtering for all four lakes is shown in Fig. 3. The reduction factors result in a seasonal filtering of the wind, with a clear tendency to curb the transfer of wind speed energy to BSIWs during the winter time, while the time series during the stratified period remains almost unchanged. Figure 3a shows the monthly averaged value of the reduction factors over the whole time series for Lake Geneva (33 years). On such average basis, both reduction factors behave similarly, however, at the event timescale, they have clearly different patterns (not shown).

Applied on the observed hourly wind speed according to Eq. (6), the filtering scheme displays a seasonal reduction of the observed values. In winter, wind intensity is reduced, whereas in summer, the time series remains rather unchanged. Results are essentially the same for Lake Constance, although filtering was globally weaker in the first months of the year. In Lake Biel, the overall filtering ratio is a smoother curve with minimal reduction occurring as early as May. In Lake Biel and Lake Neuchâtel, filtering in winter and spring is mostly driven by high lake stability, and much less by the wind duration. In particular, repeated unstratified periods, characterized by high Wedderburn numbers, lead to sharp wind filtering. In deep lakes, homogeneity occurs very infrequently and rather briefly. Especially for Lake Biel, unstable water columns appear during summer already, which leads to stronger stability-driven filtering.

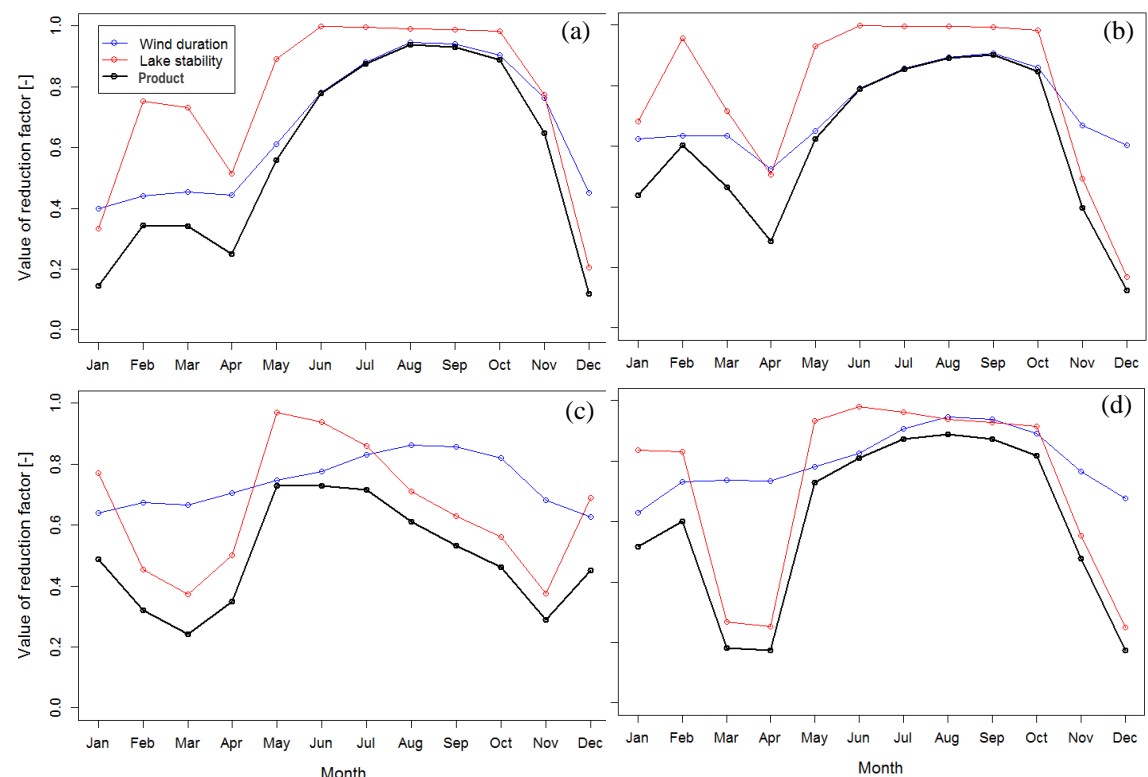

**Figure 3** – Monthly averaged reduction factors for Lake Geneva (a), Lake Constance (b), Lake Biel (c), Lake Neuchâtel (d). Shown factors: wind duration (Eq. (7): $f_{dur}$, blue), lake stability (Eq. (10): $f_{stab}$, red) and overall resulting filtering factor ($f_{dur} f_{stab}$, black).

Wind with heterogeneous direction may overall prove incapable of causing BSIWs. As a further improvement, this limitation could be parameterized with an additional factor $f_{dir}$ [-], which could be included into Eq. (6).

$f_{dir}$ can be estimated as the fraction of the time that wind direction is in the same half-circle (i.e. within $\pm\pi/2$ radians) as the average wind direction. The average wind direction is computed during each wind event identified with the method described in Section 2.2. Such filtering is expected to particularly impact systems with pronounced diurnal or seasonal wind

forcing.

### 3.2 Calibration results

Parameter calibration of the initial model for Lake Geneva yielded unrealistic parameter sets. While trying to match the observed temperatures, PEST raised $p_2$, the correction factor for the flux of latent and sensible heat, to a value exceeding 1.5. Such strong correction factors suggest that lake processes are not correctly accounted for. Hence, for all subsequent

calibration procedures, the range of calibration of $p_1$ and $p_2$ was bound between 0.80 and 1.25. The calibration results under this constraint are shown in Table 3. Forcing the lake with the filtered wind time series results in a notable improvement of $p_2$ (i.e. getting closer to 1.00). We also notice an increase of $\alpha$, which amplifies the overall magnitude of energy transfer to

seiche-induced mixing. This compensates the reduction due to the filtered wind magnitude, but increases further the impact of the long wind events (which went through the filter), in contrast to the ones that were filtered out.

These results were confirmed by applying the same procedure for Lake Constance. Notably, $p_2$, was reduced for both lakes, while $p_1$, the fit parameter controlling infrared radiation from sky, remained almost constant. One may also note that $\alpha$ is considerably higher for Lake Geneva than for Lake Constance, illustrating a strong impact of the regional wind field on BSIWs. However, the global energy transfer ratio, expressed by the expression $\alpha(f_{dur}f_{stab})^3$ (see Eq. (3)) remains rather similar in the deep lakes, pointing towards a globally efficient energy transfer: the total amount of energy transferred to BSIWs isn't much reduced by wind filtering.

| Parameter and units | Lake Geneva | | | Lake Constance | | |
| --- | --- | --- | --- | --- | --- | --- |
| | Initial model | Improved model | Relative change | Initial model | Improved model | Relative change |
| $\alpha$ [-] | 0.0709 | 0.1761 | +148 % | 0.0295 | 0.0515 | +74.6 % |
| $\alpha \cdot (f_{dur} \cdot f_{stab})^3$ [-] | 0.0709 | 0.0699 | -1.40% | 0.0295 | 0.0215 | -27.1% |
| $p_1$ [-] | 1.089 | 1.065 | -2.20 % | 1.208 | 1.177 | -2.60 % |
| $p_2$ [-] | 1.250 | 1.087 | -13.0 % | 1.064 | 0.898 | -15.6 % |
| Near-bottom RMSE [°C] | 0.32 | 0.20 | -37.5 % | 0.29 | 0.24 | -17.2 % |

Table 3 – Calibration results and near-bottom RMSE for the "deep" lakes.

Concerning Lake Neuchâtel and Lake Biel, which are both less deep with yearly overturn, we did not expect the wind filtering to affect the results significantly. Indeed, calibration showed extremely similar results (Table 4). Expectably, there was a slight increase of the $\alpha$ parameter, which compensates for the reduced wind intensity. The increase is much slighter than for the large, deep lakes, the reason of which is understood to be the lower influence of BSIWs. For Lake Neuchâtel, the value of the $p_2$ parameter dropped below the chosen upper bound, meaning that the model was not anymore forced to converge. In these shallow lakes, the global energy transfer ratio, expressed by the expression $\alpha(f_{dur}f_{stab})^3$ (see Eq. (3)), gets significantly reduced, which once more reveals that seiches are less efficiently excited there.

| Parameter and units | Lake Neuchâtel | | | Lake Biel | | |
| --- | --- | --- | --- | --- | --- | --- |
| | Initial model | Improved model | Relative change | Initial model | Improved model | Relative change |
| $\alpha$ [-] | 0.0099 | 0.0111 | +12.1 % | 0.0043 | 0.0049 | +14.0 % |
| $\alpha \cdot (f_{dur} \cdot f_{stab})^3$ [-] | 0.0099 | 0.0050 | -49.5 % | 0.0043 | 0.0016 | -62.8 % |
| $p_1$ [-] | 1.205 | 1.204 | -0.10 % | 1.095 | 1.095 | 0.00 % |
| $p_2$ [-] | 1.250 | 1.249 | -0.10 % | 1.250 | 1.250 | 0.00 % |
| Near-bottom RMSE [°C] | 0.48 | 0.47 | -2.20 % | 0.94 | 0.91 | -3.20 % |

Table 4 – Calibration results and near-bottom RMSE for the "shallow" lakes.

For the deep lakes (especially for Lake Geneva), the calibrated values of the α parameter tend to be higher than the ones found by previous studies, which ranged from 0.001 to 0.03 for large basins (Schmid and Köster, 2016; Fink et al., 2014b; Finger et al., 2007). There are several reasons that can explain this deviation:

- If the wind observed at the onshore meteorological station(s) underestimates the wind intensity over the lake, PEST is likely to compensate with a larger α value (and a larger $p_2$ value). For Lake Geneva, it has been reported that the wind in Pully may be weaker than the global wind over the lake (Lemmin and D'Adamo, 1997). The surrounding topography greatly influences the wind field over the lake and its impact on BSIWs (Valerio et al., 2017).

- The wind drag coefficient for Simstrat is given by an empirical expression (Eq. (5)) instead of a constant, in contrast to most previous studies. This leads to an overall lower drag and therefore also to a larger α value (see Eq. (3)): in our case for Lake Geneva, α increased by ~25 %.

- This is also one of the first studies where the parameter defining the fraction of short-wave radiation absorbed as heat in the uppermost layer (see Table 2) is set to a non-zero value (which was physically incorrect). This change also appears to lead to an increased α value: in our case for Lake Geneva, α increased by ~18 %.

## 3.3 Simulation results

For the following model results, the optimized set of parameters found by PEST was used. Results with filtered wind data for Lake Geneva do not improve significantly at mid-depth, even though greater variability is reproduced (Fig. 4). However, the accuracy in deep layers temperature improved greatly (Fig. 5). In accordance with Lake Geneva, the model performance improved in the deep water of Lake Constance. Reduction in root-mean-square error (RMSE) in deep water temperature substantiates these statements (see Table 3). As expected, for shallower lakes with yearly turnover, the extended model did neither improve nor worsen the results (see Table 4). Figure 6 shows the change in RMSE between the improved model and the version using unfiltered wind. In this case, positive values express a reduction of RMSE, i.e. an improvement of the model. In the deepest part of Lake Geneva, the improvement was nearly 40 %, which supports our approach.

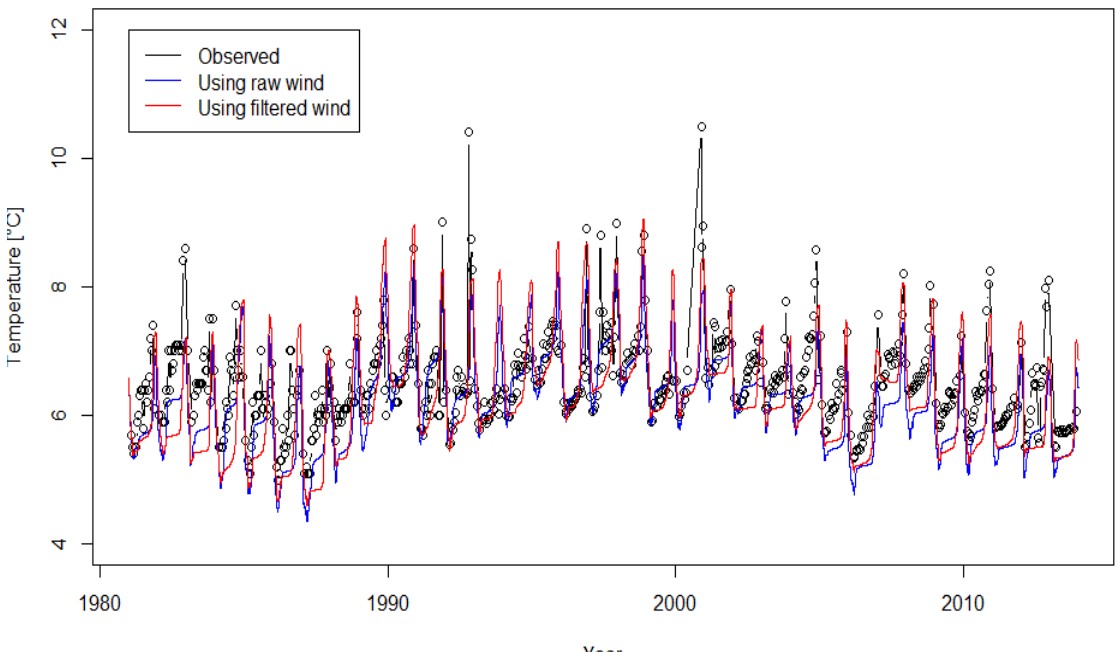

**Figure 4 – Temperature time series at 50 m depth in Lake Geneva: observations, initial model and improved model.**

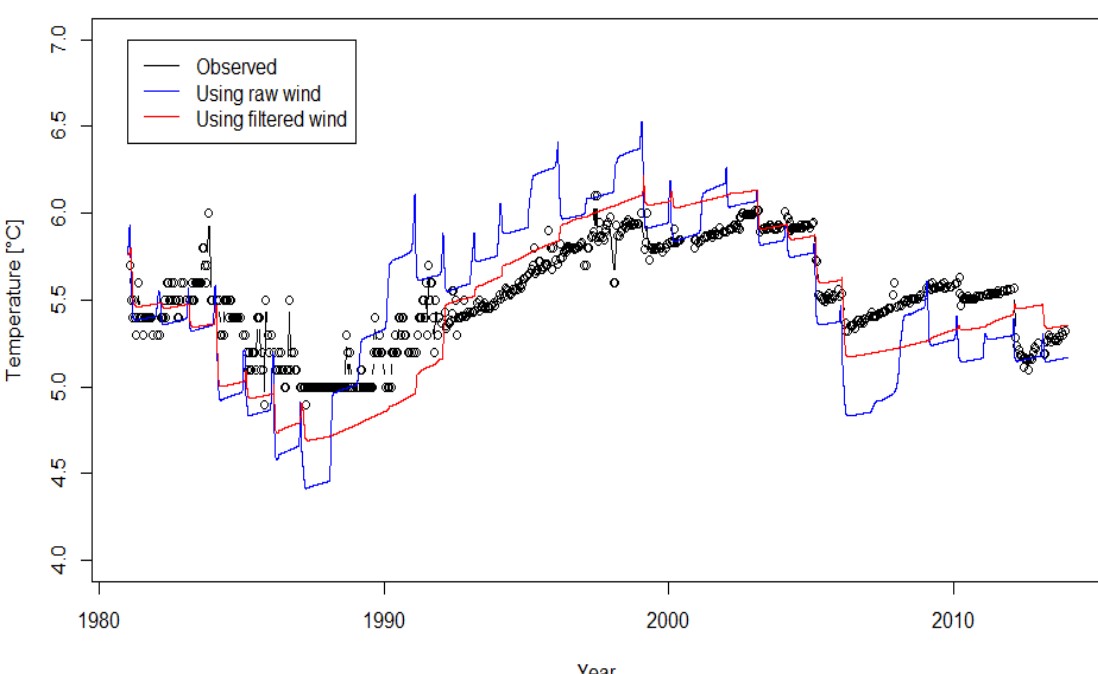

**Figure 5 – Temperature time series at 250 m depth in Lake Geneva: observations, initial model and improved model.**

5  For the shallower Lake Neuchâtel and Lake Biel, there is very little change in the results. Correcting the remaining offset, which is surely caused by other inaccuracies in the modelling schemes and in the forcing input, would require further work.

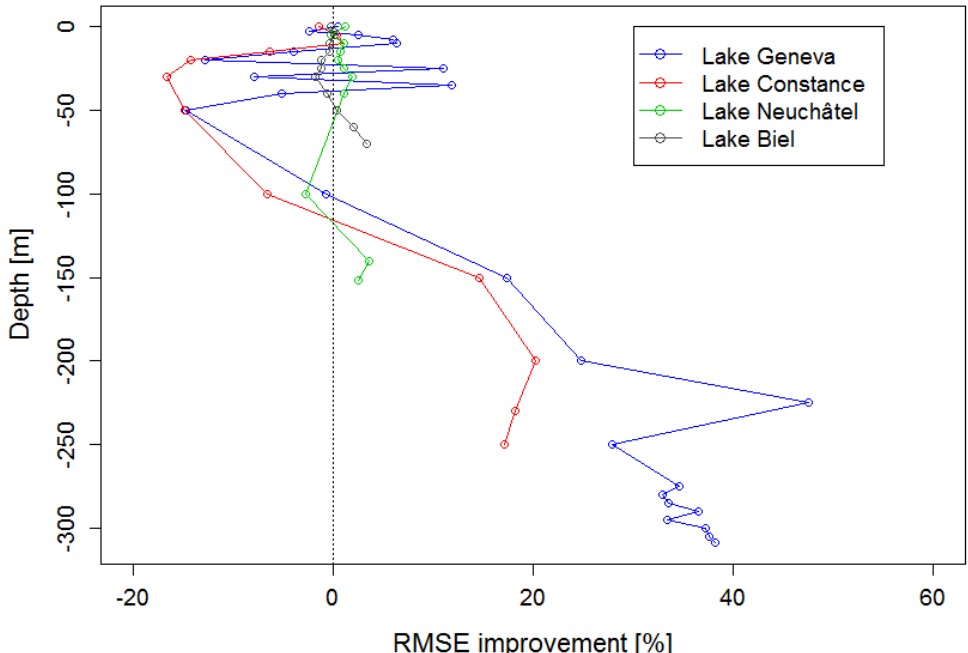

**Figure 6 – RMSE change between initial and improved model, as a function of depth. RMSE values are calculated with the temperature time series at the observations depths.**

As shown in Fig. 7 (which is an overlap of the lower pane of Fig. 2), the modelled yearly mixing depths are in much better accordance to the observed values. This is due to reduced wind-induced mixing in the hypolimnion, which then remains denser than the upper layers through the winters that are not cold enough to trigger complete mixing. Fig. 7 shows that the original version of the model largely overestimated deep convective mixing, as full lake mixing was predicted almost every year. Using wind filtering, deep seasonal mixing is more finely reproduced, which also helps towards modelling of lake-scale circulation of water, oxygen and nutrients.

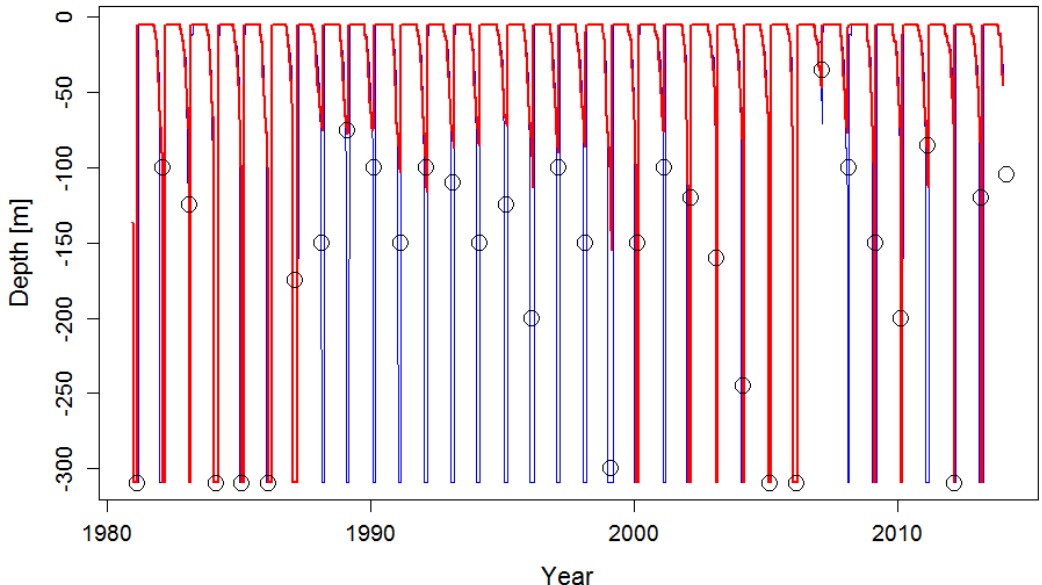

**Figure 7 – Deep mixing extent in Lake Geneva, illustrated as the mixing depth as a function of time. Circles represent observations, the curves represent corresponding model results: the original version (blue) and the version using filtered wind (red). The mixing depth is defined as the first depth at which the absolute temperature gradient exceeds 0.005 °C m$^{-1}$, using time-averaged temperature time series over 60 days.**

## 4 Conclusions

The difficulty of 1D lake models to reproduce deep water temperatures is mainly caused by the unrealistic transport of wind energy to the deeper layers of the lake. Especially in large lakes, turbulent kinetic energy produced by the dissipation of internal waves plays a major role. Our study showed that the internal wave parameterization of Simstrat can be remarkably improved for deep lakes by using filtering techniques taking into account the physical processes necessary for internal wave generation. In turn, the filters had only minor effects on shallower lakes as they fully mix over winter. However, it is very likely that our filter can be optimized for each specific case.

By considering the signal properties of the wind speed time series (notably the duration of wind events), as well as the stability of the water column, we were able to parameterize deep mixing in the basin better than with the simple parameterization provided by the original Simstrat model. This supports our assumption that BSIW processes are a key factor for the inaccuracy of 1D models in the deep hypolimnion of large lakes. Not surprisingly, the filtering had almost no effect in summer, when the Wedderburn number and the BSIW period remain small. In winter, however, filtering strongly reduced the energy transport into internal seiches. These results lead to better prediction of mixing depth and thermal structure in the deep water and open the door to a finer understanding of the process and, altogether, to a better management of lakes. Indeed, prediction of deep water reoxygenation, nutrient cycle and sediment interactions would greatly benefit from accurate modelling of deep mixing. In addition, the latter is also essential for long-term climate change studies.

Our findings emphasize the fact that the properties of complex three-dimensional processes such as BSIW formation must be better parameterized in 1D models. If such mechanisms are overly simplified, significant inaccuracies may arise. Our method has the advantages of remaining rather simple and of requiring very little computation time and very few additional parameters. In our opinion, the main drawback is the difficulty to link wind and internal water motion using only direct

approximations, especially for lakes with complicated shapes, bathymetry and wind field. For instance, we neglect the oscillatory character of BSIW, resonance and damping effects and the several possible oscillation modes. In general, there is very little research about the temporal variability of wind-induced mixing (Woolway and Simpson, 2017) and the modelling of deepwater temperature evolution could still benefit much from better understanding and parameterization of the underlying processes.

**Code and data availability**

The authors are grateful to the different institutions that provided the data used in this paper: the cantonal laboratories for the CTD profiles, the Federal Office for the Environment (FOEN) for hydrological data, the Federal Office of Meteorology and Climatology (MeteoSwiss) for meteorological data, the CIPEL for additional data on Lake Geneva and the IGKB for additional data on Lake Constance. Identification of the data is given in Table 5. The source code and documentation of the

15 numerical model, data processing and verification of the algorithms, as well as the parameter and input files can be accessed through an archived GIT repository (link: https://github.com/adrien-ga/Simstrat-BSIW/tree/v1.3, DOI: 10.5281/zenodo.841084, direct download link: https://github.com/adrien-ga/Simstrat-BSIW/archive/v1.3.zip). As previously discussed, this version of Simstrat implements the possibility to use a separate time series of wind speed for computing seiche energy, a gravity-driven intrusion algorithm for inflows and wind drag coefficient varying with wind speed.

| | CTD profiles: location ID and coordinates (CH1903) | Hydrological data: station ID(s) | Meteorological data: station short name(s) |
|---|---|---|---|
| Lake Geneva | 1696 (534700, 144950) | - | PUY |
| Lake Constance | 1644 (745450, 277120) | 2473 | GUT |
| Lake Neuchâtel | 1674 (554610, 194980) | - | NEU, FRE, CHM, CRM, MAH, MUB, PAY |
| Lake Biel | 1607 (581700, 217060) | 2085, 2307, 2446 | CRM, BIL, MUB, CHA, NEU |
| **Data providers** | **Cantonal laboratories** | **FOEN** **hydrodaten.admin.ch/en** | **MeteoSwiss** **meteoswiss.admin.ch** |

**Table 5 – Identification of the data used.**

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
