# Peer review of "Optimizing the parameterization of deep mixing and internal seiches in one-dimensional hydrodynamic models: a case study with Simstrat v1.3"

_Geoscientific Model Development, 2016_

## Short Comment (SC1) · 30 Nov 2016

Dear authors,

in my role as Executive editor of GMD, I would like to bring to your attention our Editorial version 1.1:

http://www.geosci-model-dev.net/8/3487/2015/gmd-8-3487-2015.html

This highlights some requirements of papers published in GMD, which is also available on the GMD website in the 'Manuscript Types' section:

http://www.geoscientific-model-development.net/submission/manuscript_types.html

In particular, please note that for your paper, the following requirement has not been met in the Discussions paper:

- "The main paper must give the model name and version number (or other unique identifier) in the title."

Please add a version number of Simstrat in the title upon your revised submission to GMD.

Yours,

Astrid Kerkweg

———————————————————

---

## Referee Comment (RC1) · Anonymous Referee #1 · 4 May 2017

I have reviewed this manuscript, which is generally well written albeit with a certain level of grammatical awkwardness which the copy editors will hopefully deal with.

The content of the manuscript raises some issues, however:

1. The parametrisation of energy is introduced out of context of the original model. By reading the preceding paper I am able to see where these parameters fit in, and I understand the desire by the authors to be concise and not to repeat previously published work, however the paper should also stand on its own. I suggest that a more appropriate balance in this case would be to write out the full equation set being solved

(which is not long) and then point readers at the previous paper for the details of the discretisation. This would make understanding the parametrisation presented much easier.

1.a. As a minor point, it is conventional to typeset subscripts and superscripts which are English words in upright text. f_{stab} conventionally means f_{s*t*a*b} while f_{\textrm{stab}} means what the authors intend here.

2. Verification.

There is no verification of the model: the manuscript jumps straight from explaining the parametrisation to validation (ie testing the model's effectiveness on a real problem). Given that the validation exercise involves significant calibration, it is not really possible for the reader to conclude from the paper that the implementation presented is actually correct. One would usually expect as an intermediate step, some very idealised cases where the model error is demonstrated to converge at the predicted rate. In the absence of such evidence, how is the reader supposed to conclude that the implementation is correct? Admittedly, the validation exercise provides evidence that the implementation is not disastrously wrong, but minor bugs which nonetheless affect solution quality are very common.

3. Independence of calibration and validation.

If I understand the last paragraph of page 9 correctly, the same data period was used to calibrate the models as is then used to assess the model performance. If I am mistaken in this, then the relationship between the data used for the two phases should be made much more explicit. If it is indeed the case that the same data was used for calibration and assessment, this would appear to significantly undermine the results in the paper. Using disjoint data sets for this purpose is surely necessary to demonstrate the predictive power of the model.

4. Model code availability and reproducibility.

I set out to attempt to reproduce the results in this paper using the code provided. I failed completely. There is a litany of issues:

a. Plain GitHub URL. This is not a safe archive for model code. This is for two reasons. First, the code might disappear from GitHub tomorrow, second, it is not at all clear which version of the repository was used to create the results in this paper - and as the model is developed into the future, the current version of the code at any given time will no longer be the version in this paper. Please use the GitHub Zenodo integration, or similar, to archive the precise version of the code used in this paper in a permanent location with a DOI. See: https://guides.github.com/activities/citable-code/

b. Code does not build. The provided .exe file is clearly not going to be of any use on my Linux workstation, so I set out to build the code. The user manual provides no information on how to do this. The developer manual is clearly incomplete but does at least say something.

i. The first surprise is that I need FABM. This rather important dependency isn't cited in the paper. One would rather think it would be and that what FABM provides to the model would be documented somewhere. FABM did not build for me, this may be my fault although the amount of manual intervention in the build process which the instructions appear to require makes this inherently error-prone.

ii. Apparently there is a dependency on NetCDF and HDF5, however since this section is only a header, with a single sentence body indicating that the author failed at this point, I have no indication of what I am supposed to do.

iii. I then attempted to compile kepsmodel_2016.F90 using gfortran 5.4.0, knowing that there would probably be a linker problem due to the missing libraries. However, the code does not compile due to missing keps_utilities_clean.f90. Presumably this means that keps_utilities.f90 needs to be preprocessed in some way, but no instructions to end are provided.
c. Scripts are not provided.

None of the namelists, parameter files or Matlab scripts referred to in the user manual are actually present in the repository. This despite the manuscript explicitly claiming they are there. In particular, all the code required to drive PEST appears to be absent.

d. Data is not identifiable.

Neither the paper nor the repository provides enough information for the user to obtain the data used from the original sources, since no identification is provided for the datasets. The fact that the data is not publicly available further underlines the need for verification tests based on ideal synthetic data, so that a future user can establish that the code works without depending on data to which he or she may never have access.

In short, the model code does not appear to have been published in a usable way, and the information required to reproduce the results in this paper (even assuming access to data) does not appear to be present. Indeed, the GitHub commit record appears to suggest that some code was just thrown together and committed on one day in November. It seems unlikely that the code in the repository was actually used in its current form to produce the paper. This does not conform with either the spirit or the letter of GMDs guidance on code and data availability and needs to be fixed.
* * *

---

## Author Comment (AC2) · 16 May 2017

Dear Dr. Astrid Kerkweg,

We thank you for your comment on the manuscript. As you recommend, we will add a version number of Simstrat in the title upon revised submission to GMD.

Yours,

Adrien Gaudard

---

## Referee Comment (RC2) · Anonymous Referee #2 · 4 Jun 2017

General Comments

The authors present the fairly straightforward development of a 1D model of water columns in lakes, refined to account for the extent to which wind events excite basin-scale internal waves that play a leading role in mixing at the thermocline. The Introduction covers the key physical processes (Sect. 1.1) and the modelling status quo (Sect. 1.2), justifying the current study. The Methods section (Sect. 2) is well organized, although some additional details would be helpful (see below). The Results section comprises four case studies, for a diverse range of Swiss lakes. A Conclusions section

very briefly summarizes key findings and implications. Tables and figures are clear throughout. The manuscript should be suitable for publication in GMD, subject to minor and technical revisions in response to the following comments.

Specific Comments

1. The Abstract is provides clear information of a general nature, but it could be developed to provide specific, quantitative information on the extent of improvements in accuracy of model temperatures and mixed layer depths

2. p.5, l.15: gamma depends on bottom friction and basin geometry – please add some detail on this

3. p.9, l.24: The PEST software is used to calibrate the model; beyond the reference to Doherty (2005), please define the acronym and briefly explain how PEST works

4. p.9, l.26: Two of the three parameters used in model tuning are only mentioned here; please provide details (equations?) to explain the "fit parameter for absorption of solar radiation" and the "fit parameter for the fluxes of sensible and latent heat"

5. pp. 15-16: Sect.4 provides brief conclusions; there is no explicit discussion, although brief reference to applications (p.16, lines 1-2); a more developed Discussion section would be more appropriate

Technical Corrections

1. p.3, l.20: rather then "aquatic systems", why not say "lakes"?

2. p.7, l.8: "in order to smooth the cut-off effect"

3. p.7, l.14: "both oppose excitation of BSIWs"

4. p.11, l.5: "and rather briefly"

5. p.11, lines 5-6: the sentence "A comparison of the filtering for all four lakes is shown in Fig. 3" should be moved to the start of Sect. 3.1

6. p.11, l.13: Equation (12) is hardly an equation – why is it necessary to use two different symbols for the same factor?

7. p.12, l.1: How is "average wind direction" defined?

8. p.15, l.2: "which then remains denser"

9. p.15, l.19: "In winter, however, filtering strongly . . ."

---

## Author Comment (AC3) · 13 Jun 2017

*We thank Referee #2 for the valuable comments. Our answers are given below each comment.*

Specific Comments

1. The Abstract is provides clear information of a general nature, but it could be developed to provide specific, quantitative information on the extent of improvements in accuracy of model temperatures and mixed layer depths

*We agree and will add the main quantitative results, i.e. up to 40% reduction in RMSE for the near-bottom temperature and a significantly better reproduction of the vertical extent of the seasonal deep convective mixing.*

2. p.5, l.15: gamma depends on bottom friction and basin geometry – please add some detail on this

*We will replace gamma by its equation, based on Goudsmit et al. (2002):*

$$\gamma = A_{lake} V_{lake}^{-3/2} \rho_{water}^{-1/2} C_D$$

*where $A_{lake}$ and $V_{lake}$ are the lake surface area and volume, respectively, $\rho_{water}$ the density of water and $C_D$ the bottom drag coefficient.*

3. p.9, l.24: The PEST software is used to calibrate the model; beyond the reference to Doherty (2005), please define the acronym and briefly explain how PEST works

*We will define the acronym of PEST ("Model-Independent Parameter Estimation and Uncertainty Analysis") and add the following sentence: "PEST searches for the optimal value of chosen parameters which ensure the best match between model results and available observations […]".*

4. p.9, l.26: Two of the three parameters used in model tuning are only mentioned here; please provide details (equations?) to explain the "fit parameter for absorption of solar radiation" and the "fit parameter for the fluxes of sensible and latent heat"

*We will give more details about these two parameters as follows, based on Goudsmit et al. (2002):*

- *$p_1$ (fit parameter for absorption of longwave radiation) linearly scales the amount of heat that is absorbed in the lake water from the incoming longwave radiation from the atmosphere;*

- *$p_2$ (fit parameter for the fluxes of sensible and latent heat) linearly scales the exchange of sensible and latent heat between the lake surface and the atmosphere.*

*We will explain the significance of these two parameters as follows: "$p_1$ and $p_2$ account for the fact that, in specific cases, there is always a difference between the heat flux formulas and the effective fluxes."*

5. pp. 15-16: Sect.4 provides brief conclusions; there is no explicit discussion, although brief reference to applications (p.16, lines 1-2); a more developed Discussion section would be more appropriate

*We agree, however we purposely kept the manuscript as concise as possible. We will develop the following items in more detail:*

- *We will further comment Figure 7, as follows: "Fig. 7 shows that the original version of the model largely overestimated deep convective mixing, as full lake mixing was being predicted almost every year. Using wind filtering, deep seasonal mixing is more finely*

*reproduced, which also helps towards modelling of lake-scale circulation of water, oxygen and nutrients."*

- *We will also discuss our calibrated values for the α parameter for the deep lakes, which tend to be higher than the ones found by previous studies. For example, regarding Lake Geneva, one reason is that the meteorological station chosen for the study underestimates the wind intensity over the lake, thereby leading to larger α values.*

1. p.3, l.20: rather then "aquatic systems", why not say "lakes"?

   *We will replace the expression by "lakes and reservoirs".*

2. p.7, l.8: "in order to smooth the cut-off effect"

3. p.7, l.14: "both oppose excitation of BSIWs"

4. p.11, l.5: "and rather briefly"

5. p.11, lines 5-6: the sentence "A comparison of the filtering for all four lakes is shown in Fig. 3" should be moved to the start of Sect. 3.1

6. p.11, l.13: Equation (12) is hardly an equation – why is it necessary to use two different symbols for the same factor?

7. p.12, l.1: How is "average wind direction" defined?

8. p.15, l.2: "which then remains denser"

9. p.15, l.19: "In winter, however, filtering strongly …"

   *We thank Referee #2 for these technical corrections and will change the manuscript accordingly. In particular, we will remove Equation (12) and instead explain how average wind direction can be calculated.*

---

## Author Response (AR1)

**Author's response**

*NB: the mentioned pages and lines refer to the marked-up manuscript.*

**Referee #1**

1. The parametrisation of energy is introduced out of context of the original model. By reading the preceding paper I am able to see where these parameters fit in, and I understand the desire by the authors to be concise and not to repeat previously published work, however the paper should also stand on its own. I suggest that a more appropriate balance in this case would be to write out the full equation set being solved (which is not long) and then point readers at the previous paper for the details of the discretisation. This would make understanding the parametrisation presented much easier.

> *We agree with Referee #1 that the context of the new parametrisation was insufficiently explained in the manuscript, and that it is important that the present paper stands on its own. However, we think that presenting the full set of equations, which would then also require explaining the full set of parameters used in these equations, would distract too much from the main focus of the manuscript. Both are fully developed in Goudsmit et al. 2002. Instead, we added a more detailed explanation about the meaning and relevance of the modified parameter ($P_{seiche}$) in the Simstrat model, as follows: "Within the standard system of equations of a k-ε model, where k is the turbulent kinetic energy and ε its dissipation, $P_{seiche}$ is used as a source term in the differential equation for k and, multiplied by ε/k and a constant, as a source term in the differential equation for ε." on page 5, lines 15-17.*

1.a. As a minor point, it is conventional to typeset subscripts and superscripts which are English words in upright text. f_{stab} conventionally means f_{s*t*a*b} while f_{\textrm{stab}} means what the authors intend here.

> *We agree and changed the syntax accordingly throughout the manuscript.*

2. Verification. There is no verification of the model: the manuscript jumps straight from explaining the parametrisation to validation (ie testing the model's effectiveness on a real problem). Given that the validation exercise involves significant calibration, it is not really possible for the reader to conclude from the paper that the implementation presented is actually correct. One would usually expect as an intermediate step, some very idealised cases where the model error is demonstrated to converge at the predicted rate. In the absence of such evidence, how is the reader supposed to conclude that the implementation is correct? Admittedly, the validation exercise provides evidence that the implementation is not disastrously wrong, but minor bugs which nonetheless affect solution quality are very common.

> *We agree with Referee #1 that model verification would be useful to prevent the occurrence of bugs. We propose adding an idealized case where wind is a periodic rectangular function of variable frequency affecting a two-layer basin. We can then show that, depending on the frequency of the wind function, filtering is correctly performed and transmitted to the model and that BSIW excitation occurs as expected. There is however to our knowledge no analytical solution or other method that could predict the exact output of the model running such an idealized case.*

> *We added such an idealized case on the GitHub (folder: "Simstrat_WindFiltering/", parameter file: "kepsilon_IdealizedCase.par", input files: "IdealizedCase/", results:*

*"IdealizedCase_Results/" and "IdealizedCase_Results_wfilt/"). The central features of the setup are:*

- *Basin with a depth of 100 m and a surface area of 100 km$^2$ (from surface to bottom).*
- *Temperature of 15°C from 0 to 25 m depth, temperature of 5°C below 25 m depth.*
- *No inflows and outflows and no heat flux at the surface, therefore there are only internal water and heat exchanges.*
- *Wind forcing (5 m s$^{-1}$ in both directions) as a rectangular function with period of either 1 hour (hereafter W1; file "Forcing_period1.dat3") or 24 hours (hereafter W24; file "Forcing_period24.dat3").*
- *Default set of parameters used for all model runs.*

*The seiching period of the first mode of the basin is around 27 hours (Equation (1) of the manuscript). Although the overall wind average is the same, W24 would result in a stronger response of BSIWs within the idealized basin than W1, as the wind period is much closer to the seiching period of the basin.*

*This idealized case allows to verify the following aspects of our algorithms:*

- *If there is no wind forcing, the temperature evolution in the idealized basin is governed solely by molecular diffusion. Wind promotes faster mixing between the layers and progressively drives the "thermocline" deeper.*
- *W24 induces an increased excitation of the basin when compared to W1 (e.g., clear temperature oscillations at the interface at 25 m depth).*
- *Wind filtering (script "Simstrat_WindFiltering.R") reduces the average intensity of W1 by ~37 % and of W24 by ~1 %. As a result, the filtered W24 excites the idealized basin significantly more efficiently than the filtered W1.*

*After thorough discussion, we decided not to add this test case to the manuscript. We feel that the information that can be gained by including it does not outweigh the length and complexity that it adds to the manuscript. We then prefer to keep the manuscript focused on real systems.*

3. Independence of calibration and validation. If I understand the last paragraph of page 9 correctly, the same data period was used to calibrate the models as is then used to assess the model performance. If I am mistaken in this, then the relationship between the data used for the two phases should be made much more explicit. If it is indeed the case that the same data was used for calibration and assessment, this would appear to significantly undermine the results in the paper. Using disjoint data sets for this purpose is surely necessary to demonstrate the predictive power of the model.

*Our work aims at explaining the importance of an overlooked physical process and including it into the model. In our case, we propose external processing of the wind time series to improve BSIW parametrisation. We therefore want to highlight the improvement that the proposed modification brings in comparison to the original parametrisation, rather than demonstrate the performance and predictive power of a new model. The single difference between the models is that the "modified version" takes a different time series of wind as input into Equation (3). As a measure of the improvement, we compare the results for the new model version with those for the original model version. Both model versions are calibrated in exactly the same way. Because the revised model does not include any additional calibration parameters compared to the original version, we don't think that separating the data in a calibration and a validation*

*period would yield additional information concerning the usefulness of the new parametrisation.*

4. Model code availability and reproducibility.

I set out to attempt to reproduce the results in this paper using the code provided. I failed completely. There is a litany of issues:

a. Plain GitHub URL. This is not a safe archive for model code. This is for two reasons. First, the code might disappear from GitHub tomorrow, second, it is not at all clear which version of the repository was used to create the results in this paper - and as the model is developed into the future, the current version of the code at any given time will no longer be the version in this paper. Please use the GitHub Zenodo integration, or similar, to archive the precise version of the code used in this paper in a permanent location with a DOI. See: https://guides.github.com/activities/citable-code/

*We agree that there must be a safe archive for model code and related files. Simstrat was only recently transferred to version control. As suggested, we archived the code in a permanent location with a DOI. The link to this archive is given in the "Code and data availability" Section of the manuscript, on page 18, lines 18-19.*

b. Code does not build. The provided .exe file is clearly not going to be of any use on my Linux workstation, so I set out to build the code. The user manual provides no information on how to do this. The developer manual is clearly incomplete but does at least say something.

i. The first surprise is that I need FABM. This rather important dependency isn't cited in the paper. One would rather think it would be and that what FABM provides to the model would be documented somewhere. FABM did not build for me, this may be my fault although the amount of manual intervention in the build process which the instructions appear to require makes this inherently error-prone.

*We address this comment in the next answer.*

ii. Apparently there is a dependency on NetCDF and HDF5, however since this section is only a header, with a single sentence body indicating that the author failed at this point, I have no indication of what I am supposed to do.

*The manual provided on the GitHub, although it is not a final version, covers a larger scope than required for the model used in the manuscript. Neither FABM, nor NetCDF nor HDF5 are required to compile the code provided on the GitHub or run the simulations presented in the manuscript. We are sorry if this was not clear and clarified the manual accordingly. We updated the Readme file on the GitHub with simple instructions on how to run the model and visualize the results.*

iii. I then attempted to compile kepsmodel_2016.F90 using gfortran 5.4.0, knowing that there would probably be a linker problem due to the missing libraries. However, the code does not compile due to missing keps_utilities_clean.f90. Presumably this means that keps_utilities.f90 needs to be preprocessed in some way, but no instructions to end are provided.

*Using gfortran, compilation of the source code should work independently of the platform, as no extra library is required. However, as Referee #1 rightly highlighted, a typo prevented compilation: at the end of the main file kepsmodel_2016.f90, the suffix "_clean" should be removed from the two include filenames. We corrected the code accordingly.*

c. Scripts are not provided. None of the namelists, parameter files or Matlab scripts referred to in the user manual are actually present in the repository. This despite the manuscript explicitly claiming they are there. In particular, all the code required to drive PEST appears to be absent.

> *For the sake of simplicity, we decided not to provide the files that were not strictly needed to compile the model. However, we agree that the parameter files and the processing scripts should be provided, and we added them to the repository. We also provided the code used to run calibration through PEST, and updated the Readme file on the GitHub with simple instructions on how to calibrate the model.*

d. Data is not identifiable. Neither the paper nor the repository provides enough information for the user to obtain the data used from the original sources, since no identification is provided for the datasets. The fact that the data is not publicly available further underlines the need for verification tests based on ideal synthetic data, so that a future user can establish that the code works without depending on data to which he or she may never have access. In short, the model code does not appear to have been published in a usable way, and the information required to reproduce the results in this paper (even assuming access to data) does not appear to be present. Indeed, the GitHub commit record appears to suggest that some code was just thrown together and committed on one day in November. It seems unlikely that the code in the repository was actually used in ist current form to produce the paper. This does not conform with either the spirit or the letter of GMDs guidance on code and data availability and needs to be fixed.

> *We agree and provided information allowing better identification of the data. For clearness, we removed the incomplete references from page 9, lines 4, 11 and 12-13. We added full identification of the datasets in the "Code and data availability" Section, on page 18, lines 11-15 and in Table 5. In addition, in order to facilitate independent testing of the model, we provided an excerpt of the data we used on the GitHub. As the authors are not the data owners, we don't have the right to make the complete datasets used in the model freely available. These are available upon request from the data owners listed in the "Code and data availability" Section of the manuscript, on page 18, lines 11-15.*

**Referee #2**

Specific Comments

1. The Abstract is provides clear information of a general nature, but it could be developed to provide specific, quantitative information on the extent of improvements in accuracy of model temperatures and mixed layer depths

> *We agree and added the main quantitative results, i.e. up to 40 % reduction in RMSE for the near-bottom temperature and a significantly better reproduction of the vertical extent of the seasonal deep convective mixing, on page 1, lines 16-18.*

2. p.5, l.15: gamma depends on bottom friction and basin geometry – please add some detail on this

> *We replaced gamma by its equation, based on Goudsmit et al. (2002), on page 5, lines 21-23:*

$$\gamma = A_{lake} V_{lake}^{-3/2} \rho_{water}^{-1/2} C_{Deff}$$

> *where $A_{lake}$ and $V_{lake}$ are the lake surface area and volume, respectively, $\rho_{water}$ the density of water and $C_{Deff}$ the bottom drag coefficient.*

3. p.9, l.24: The PEST software is used to calibrate the model; beyond the reference to Doherty (2005), please define the acronym and briefly explain how PEST works

*We defined the acronym of PEST ("Model-Independent Parameter Estimation and Uncertainty Analysis") and added the following sentence: "PEST searches for the optimal value of chosen parameters which ensure the best match between model results and available observations (here, temperature measurements over the simulation period)." on page 10, lines 4-7. In addition, the code to run calibration through PEST is provided on the GitHub repository.*

4. p.9, l.26: Two of the three parameters used in model tuning are only mentioned here; please provide details (equations?) to explain the "fit parameter for absorption of solar radiation" and the "fit parameter for the fluxes of sensible and latent heat"

*We added more details about these two parameters as follows, based on Goudsmit et al. (2002), on page 10, lines 9-11:*

- *$p_1$ (fit parameter for absorption of longwave radiation) linearly scales the amount of heat that is absorbed in the lake water from the incoming longwave radiation from the atmosphere.*

- *$p_2$ (fit parameter for the fluxes of sensible and latent heat) linearly scales the exchange of sensible and latent heat between the lake surface and the atmosphere.*

*We explained the significance of these two parameters as follows: "$p_1$ and $p_2$ account for the fact that, in specific cases, there is always a difference between the heat flux formulas and the effective fluxes, as the formulas are empirical and the meteorological data is not always fully representative for the lake surface." on page 10, lines 11-13.*

5. pp. 15-16: Sect.4 provides brief conclusions; there is no explicit discussion, although brief reference to applications (p.16, lines 1-2); a more developed Discussion section would be more appropriate

*We agree, however we purposely kept the manuscript as concise as possible. We developed the following items in more detail:*

- *We further commented Figure 7, as follows: "Fig. 7 shows that the original version of the model largely overestimated deep convective mixing, as full lake mixing was being predicted almost every winter. Using wind filtering, deep seasonal mixing is more finely reproduced, which also helps towards modelling of lake-scale circulation of water, oxygen and nutrients." on page 16, lines 6-9.*

- *We added a discussion about our calibrated values for the α parameter for the deep lakes, which tend to be higher than the ones found by previous studies, on page 14, lines 1-13.*

- *We added a concluding sentence about possible further research on page 18, lines 6-9.*

1. p.3, l.20: rather then "aquatic systems", why not say "lakes"?

*We replaced the expression by "lakes and reservoirs" on page 4, line 3.*

2. p.7, l.8: "in order to smooth the cut-off effect"

*Corrected on page 7, line 16.*

3. p.7, l.14: "both oppose excitation of BSIWs"

*Corrected on page 7, line 22.*

4. p.11, l.5: "and rather briefly"

*Corrected on page 11, line 18.*

5. p.11, lines 5-6: the sentence "A comparison of the filtering for all four lakes is shown in Fig. 3" should be moved to the start of Sect. 3.1

*Moved accordingly to page 11, line 7.*

6. p.11, l.13: Equation (12) is hardly an equation – why is it necessary to use two different symbols for the same factor?

*We removed Equation (12).*

7. p.12, l.1: How is "average wind direction" defined?

*We defined how average wind direction can be calculated on page 12, lines 8-10.*

8. p.15, l.2: "which then remains denser"

*Corrected on page 16, line 6.*

9. p.15, l.19: "In winter, however, filtering strongly …"

*Corrected on page 17, line 17.*

**Other changes**

- *It is inexact to compare the calibrated $\alpha$ values of the initial and improved models. The global filtering $f$ multiplies the wind speed used in Equation (3), where it is then raised to the cube. Therefore, the overall filtering can be expressed as follows:*

  *Initial model* $\qquad P_{seiche} = \alpha A_{lake} \rho_{air} C_{10} U_{10}^3$

  *Improved model* $\qquad P_{seiche} = \alpha f^3 A_{lake} \rho_{air} C_{10} U_{10}^3$

  *A more relevant comparison is then $\alpha$ of the initial model with $\alpha f^3$ of the improved model. We adapted Tables 3 and 4 accordingly on page 13 and briefly described and discussed this expression and this new comparison on page 13, lines 8-10 and 17-18.*

- *We added references to four recent relevant studies: Schwefel et al. (2016), Woolway and Simpson (2017), Fink et al. (2016), Valerio et al. (2017).*

- *We added more references to Table 1 and to the "Code and data availability" Section on page 8, line 22 and page 9, lines 19 and 26.*

- *We made several other minor corrections, clarifications and referencing changes.*

[revised manuscript text omitted]

Råman Vinnå, L., Wüest, A., Bouffard, D.: Physical effects of thermal pollution in lakes, Water Resour. Res. 53, 3968–3987, 2017.

Read, J. S., Hamilton, D. P., Desai, A. R., Rose, K. C., MacIntyre, S., Lenters, J. D., Smyth, R. L., Hanson, P. C., Cole, J. J., Staehr, P. A., Rusak, J. A., Pierson, D. C., Brookes, J. D., Laas, A., Wu, C. H.: Lake-size dependency of wind shear and convection as controls on gas exchange, Geophys. Res. Lett. 39, 9, L09405, 2012.

35

Schmid, M., Köster, O.: Excess warming of a Central European lake driven by solar brightening, Water Resour. Res. 52, 10, 8103–8116, 2016.

Schwefel, R., Gaudard, A., Wüest, A., Bouffard, D.: Effects of climate change on deep-water oxygen and winter mixing in a deep lake (Lake Geneva)—Comparing observational findings and modeling, Water Resour. Res. 52, 11, 8811–8826, 2016.

40

Shimoda, Y., Azim, M. E., Perhar, G., Ramin, M., Kenney, M. A., Sadraddini, S., Gudimov, A., Arhonditsis, G. B.: Our current understanding of lake ecosystem response to climate change: What have we really learned from the north temperate deep lakes?, J. Gt. Lakes Res. 37, 1, 173–193, 2011.

Shintani, T., de la Fuente, A., Niño, Y., Imberger, J.: Generalizations of the Wedderburn number: Parameterizing upwelling in stratified lakes, Limnol. Oceanogr. 55, 3, 1377–1389, 2010.

Stepanenko, V. M., Goyette, S., Martynov, A., Perroud, M., Fang, X., Mironov, D.: First steps of a lake model intercomparison Project: LakemiP, Boreal Environ. Res. 15, 191–202, 2010.

Stepanenko, V. M., Jöhnk, K. D., Machulskaya, E., Perroud, M., Subin, Z., Nordbo, A., Mammarella, I., Mironov, D.: Simulation of surface energy fluxes and stratification of a small boreal lake by a set of one-dimensional models, Tellus A 66, 1, 21389, 2014.

Stepanenko, V. M., Mammarella, I., Ojala, A., Miettinen, H., Lykosov, V., Vesala, T.: LAKE 2.0: a model for temperature, methane, carbon dioxide and oxygen dynamics in lakes, Geosci. Model Dev. 9, 5, 1977–2006, 2016.

Stepanenko, V. M., Martynov, A., Jöhnk, K. D., Subin, Z. M., Perroud, M., Fang, X., Beyrich, F., Mironov, D., Goyette, S.: A one-dimensional model intercomparison study of thermal regime of a shallow, turbid midlatitude lake, Geosci. Model Dev. 6, 4, 1337–1352, 2013.

Stevens, C., Imberger, J.: The initial response of a stratified lake to a surface shear stress, J. Fluid Mech. 312, 39–66, 1996.

Straile, D., Kerimoglu, O., Peeters, F., Jochimsen, M. C., Kümmerlin, R., Rinke, K., Rothhaupt, K. O.: Effects of a half a millennium winter on a deep lake – a shape of things to come?, Glob. Change Biol. 16, 10, 2844–2856, 2010.

Straile, D., Livingstone, D. M., Weyhenmeyer, G. A., George, D. G.: The response of freshwater ecosystems to climate variability associated with the North Atlantic Oscillation. Wiley Online Library, 2003.

Thiery, W., Stepanenko, V. M., Fang, X., Jöhnk, K. D., Li, Z., Martynov, A., Perroud, M., Subin, Z. M., Darchambeau, F., Mironov, D., Van Lipzig, N. P. M.: LakeMIP Kivu: evaluating the representation of a large, deep tropical lake by a set of one-dimensional lake models, Tellus, Ser. A, 66, 21390, 2014.

Thorpe, S. A., Lemmin, U., Perrinjaquet, C., Fer, I.: Observations of the thermal structure of a lake using a submarine, Limnol. Oceanogr. 44, 1575–1582, 1999.

Toffolon, M., Rizzi, G.: Effects of spatial wind inhomogeneity and turbulence anisotropy on circulation in an elongated basin: A simplified analytical solution, Adv. Water Resour. 32, 10, 1554–1566, 2009.

Umlauf, L., Lemmin, U.: Interbasin exchange and mixing in the hypolimnion of a large lake: The role of long internal waves, Limnol. Oceanogr. 50, 5, 1601–1611, 2005.

Valipour, R., Bouffard, D., Boegman, L., Rao, Y. R.: Near-inertial waves in Lake Erie, Limnol. Oceanogr. 60, 1522–1535, 2015.

Wang, W., Roulet, N. T., Strachan, I. B., Tremblay, A.: Modeling surface energy fluxes and thermal dynamics of a seasonally ice-covered hydroelectric reservoir, Sci. Total Environ. 550, 793–805, 2016.

Wiegand, R. C., Carmack, E. C.: The climatology of internal waves in a deep temperate lake, J. Geophys. Res. Oceans 91, C3, 3951–3958, 1986.

Wood, T. M., Wherry, S. A., Piccolroaz, S., Girdner, S. F.: Simulation of deep ventilation in Crater Lake, Oregon, 1951–2099. US Geological Survey, 2016.

Woolway, R. I., Simpson, J. H.: Energy input and dissipation in a temperate lake during the spring transition, Ocean Dyn. 1–13, 2017.

Wüest, A., Lorke, A.: Small-scale hydrodynamics in lakes, Annu. Rev. Fluid Mech. 35, 1, 373–412, 2003.